# Hereditary Renal Cancer Syndromes

**DOI:** 10.3390/medsci12010012

**Published:** 2024-02-18

**Authors:** Grigory A. Yanus, Ekaterina Sh. Kuligina, Evgeny N. Imyanitov

**Affiliations:** 1Department of Medical Genetics, Saint-Petersburg State Pediatric Medical University, 194100 Saint-Petersburg, Russia; octavedoctor@yandex.ru; 2Department of Tumor Growth Biology, N.N. Petrov National Medical Research Center of Oncology, 197758 Saint-Petersburg, Russia; kate.kuligina@gmail.com; 3Laboratory of Molecular Biology, Kurchatov Complex for Medical Primatology, National Research Centre “Kurchatov Institute”, 354376 Sochi, Russia

**Keywords:** kidney cancer, hereditary cancer syndromes, von Hippel–Lindau disease, Birt–Hogg–Dubé syndrome, familial papillary renal cell carcinoma, hereditary leiomyomatosis and renal cell cancer, tuberous sclerosis, next-generation sequencing, targeted therapy, belzutifan

## Abstract

Familial kidney tumors represent a rare variety of hereditary cancer syndromes, although systematic gene sequencing studies revealed that as many as 5% of renal cell carcinomas (RCCs) are associated with germline pathogenic variants (PVs). Most instances of RCC predisposition are attributed to the loss-of-function mutations in tumor suppressor genes, which drive the malignant progression via somatic inactivation of the remaining allele. These syndromes almost always have extrarenal manifestations, for example, von Hippel–Lindau (VHL) disease, fumarate hydratase tumor predisposition syndrome (FHTPS), Birt–Hogg–Dubé (BHD) syndrome, tuberous sclerosis (TS), etc. In contrast to the above conditions, hereditary papillary renal cell carcinoma syndrome (HPRCC) is caused by activating mutations in the *MET* oncogene and affects only the kidneys. Recent years have been characterized by remarkable progress in the development of targeted therapies for hereditary RCCs. The HIF2aplha inhibitor belzutifan demonstrated high clinical efficacy towards VHL-associated RCCs. mTOR downregulation provides significant benefits to patients with tuberous sclerosis. MET inhibitors hold promise for the treatment of HPRCC. Systematic gene sequencing studies have the potential to identify novel RCC-predisposing genes, especially when applied to yet unstudied populations.

## 1. Introduction

Hereditary cancer syndromes (HCSs) are the most common category of Mendelian genetic diseases in humans. Clinical and genomic data indicate that approximately 2–3% of people carry germline genetic variants, which render a 50–100% lifetime probability of developing a tumor in a particular organ or tissue. This risk may be an order or a few orders of magnitude higher than in the general population. *BRCA1/2*-driven breast–ovarian cancer syndrome and hereditary non-polyposis colorectal cancer (HNPCC, also known as Lynch syndrome) are relatively common diseases affecting approximately 1 out of 200–300 people each. There are a few dozen other types of HCSs; however, they have a significantly lower incidence. Indeed, the knowledge on most of the HCS types is limited to the description of several thousand or even several hundred affected individuals with many nuances remaining poorly understood [1,2].

Hereditary cancer syndromes are associated with the presence of a germline defect in a cancer-predisposing gene. The terms “hereditary mutation” or “germline mutation” were commonly utilized in medical genetics until recently; modern literature tends to use “pathogenic variant” (PV) or “pathogenic allele” instead. Indeed, the word “mutation” means the current event, which results in changes in the original nucleotide sequence; therefore, it is more applicable to various experimental models, the description of evolution, the analysis of somatic alterations associated with the process of malignant transformation, etc. Instead, the majority of germline allelic variations emerged many centuries ago, and they are just being transmitted through generations without involving novel events. Still, many specialists continue to describe highly penetrant, disease-causing alleles as mutations, assuming that this definition is well-recognized in the clinical context [3].

The majority of HCSs are autosomal-dominant diseases. In the most common scenario, a cancer-associated PV, being acquired from one of the parents or emerging de novo, is present in every cell of the body but does not lead to phenotypic consequences due to the presence of the second allele of the same gene. The cancer development is triggered by the somatic inactivation of the remaining gene copy, be it a deletion (loss of heterozygosity, LOH), a point mutation, or epigenetic silencing. This two-hit mechanism, initially predicted by Alfred G. Knudson, is observed in the majority of HCSs, although some other roots have been described as well [1,2,4].

HCSs involving the kidneys are significantly less common when compared to hereditary breast–ovarian or colorectal cancer syndromes. They include approximately a dozen different conditions, which make a noticeable impact on kidney cancer morbidity when considered together [5,6,7,8,9,10,11,12,13,14,15,16,17,18,19,20,21,22,23,24,25,26,27,28,29,30,31,32,33,34,35,36,37,38,39,40,41,42,43,44,45,46,47,48,49,50,51,52,53,54,55,56,57,58,59,60,61,62,63,64,65,66,67,68,69,70]. The majority of kidney-related HCSs are not limited to renal malignancies but also affect other organs. The most well-known renal HCSs are von Hippel–Lindau (VHL) disease, fumarate hydratase tumor predisposition syndrome (FHTPS; previously known as hereditary leiomyomatosis and renal cell carcinoma (HLRCC)), and hereditary papillary renal cell carcinoma syndrome (HPRCC). In addition, kidney involvement is often observed in Birt–Hogg–Dubé syndrome, tuberous sclerosis, *BAP1* tumor predisposition syndrome, etc. (Figure 1, Table 1 and Table 2). Some data indicate that Cowden syndrome, Lynch syndrome, *CHEK2* germline mutations, etc., may also be associated with an increased risk of renal malignancies [9,71,72]. This review provides an update on clinical and genetic aspects of inherited predisposition to kidney cancer.

## 2. Hereditary Conditions Associated with Increased Risk of Renal Malignancies

### 2.1. Von Hippel–Lindau Syndrome

Von Hippel–Lindau syndrome (VHL) was independently described by German ophthalmologist Eugen von Hippel in the year 1904 and Swedish pathologist Arvid Lindau in the year 1927. It is manifested mainly by hemangioblastomas, clear cell renal cell carcinomas (ccRCCs), and paragangliomas (pheochromocytomas), although some other benign and malignant lesions, particularly pancreatic tumors, may be observed as well. The estimates of the population prevalence of VHL vary between 1:36,000 and 1:91,000. In fact, relevant epidemiological studies were carried out in only a few countries, and worldwide VHL statistics remains to be obtained [6]. *VHL* pathogenic variants contribute to 0.3–3% of renal cell cancer (RCC) incidence. This syndrome is an autosomal-dominant disease that is caused by the inheritance of pathogenic variants in the *VHL* gene. These mutations are either represented by protein-truncating variants, which result in complete inactivation of the gene, or amino acid substitutions, frequently rendering only partial alteration of the function of the corresponding protein [73].

The spectrum of VHL-associated tumors correlates with the character of the *VHL* mutation and the degree of its inactivating defect; therefore, the disease is subdivided into types 1, 2A, 2B, and 2C. *VHL* type 1 pathogenic alleles strongly affect VHL protein function and are associated with a relatively low risk of paragangliomas. *VHL* type 2 PVs are missense mutations, which result in less severe alteration of VHL properties and render a high probability of paraganglioma development. RCC occurrence is characteristic, mainly for VHL types 1 and 2B. VHL type 2A families have instances of both hemangioblastomas and paragangliomas, but a relatively low incidence of RCC, while VHL type 2C is associated with paragangliomas only. There are some functional differences between *VHL* type 2A, 2B, and 2C missense variants [20,74,75]. This classification mainly reflects a history of clinical and academic VHL research; however, the differences in the spectrum of associated tumors are not clear-cut and cannot be currently used for the planning of medical surveillance for *VHL* PV carriers [6].

VHL is a relatively small protein, which is located on chromosome 3p25.3 and consists of 213 amino acids encoded by three exons. There are no other genes that cause the VHL-associated phenotype; therefore, patients with suspected VHL syndrome, in theory, do not require multigene testing. Right after the discovery of the *VHL* gene in the year 1993, its testing relied on the analysis of gross deletions by Southern blot coupled with conventional Sanger sequencing [76]. The Southern blot technique was later replaced by multiplex ligation-dependent probe amplification (MLPA). So-called large gene rearrangements (LGRs), i.e., losses or insertions of one or several exons, account for 10–20% of *VHL* pathogenic alleles [54,55,77]. For the time being, most patients with suspected hereditary disease are subjected to multigene next-generation sequencing (NGS). This renders some risk of under-diagnosis for VHL patients, as NGS is insufficiently validated for the detection of LGRs [78,79].

Protein-truncating *VHL* pathogenic alleles appear to have approximately complete penetrance. On the opposite end, *VHL* type 2C missense mutations have only borderline influence on the protein function and, therefore, may not be associated with severe phenotypic consequences; it is likely that their carriers are missed when rigorous clinical criteria are applied towards VHL genetic testing [20,73,75]. Indeed, the data obtained from exome sequencing databases suggest that the population frequency of *VHL* gene heterozygotes clearly exceeds the one for diagnosed VHL patients. Some studies revealed the persistence of founder *VHL* mutations in some populations; however, their proportion among the entire spectrum of *VHL* genetic lesions is relatively low [80,81]. Up to 20% of *VHL* pathogenic variants emerge de novo [5].

VHL-associated RCCs usually arise in the background of multiple renal cysts with a median age at cancer diagnosis approaching 39 years. RCCs are often multiple and bilateral; however, they tend to have indolent growth. The “watch-and-wait” approach is usually applied to renal masses sized below 3 cm in diameter, and nephron-sparing techniques are considered for tumor ablation [82] (Table 3).

Stereotactic radiotherapy may be utilized for organ-preserving destruction of malignant lesions instead of surgery [91].

VHL is an E3 ubiquitin ligase, which is involved in oxygen sensing and regulates the degradation of hypoxia-inducible factors. Somatic *VHL* inactivation results in the accumulation of HIF (hypoxia-inducible factor) and, consequently, stimulation of tumor angiogenesis. The phenotype related to HIF activation in the presence of a normal oxygen supply is often called “pseudohypoxia” [92]. There are multiple other functions of VHL protein, which are not directly related to HIF regulation; however, their role in carcinogenesis is less studied. Upregulation of HIF2alpha is particularly important for RCC pathogenesis. Somatic inactivation of *VHL* is characteristic both for hereditary and sporadic RCCs; therefore, the majority of RCCs are characterized by massive and abnormal vasculature. This explains the significant efficacy of bevacizumab, an antibody derived against vascular endothelial growth factor A, in the RCC therapy (Table 4). In addition, a number of small-molecule multikinase inhibitors (sunitinib, axitinib, pazopanib, cabozantinib, etc.), which are often marketed as antiangiogenic drugs, have been incorporated into the treatment standards for metastatic RCCs [93]. The development of belzutifan (previously called MK-6482 or PT2977), an allosteric inhibitor of HIF2alpha, led to a breakthrough in the treatment of RCCs arising in *VHL* germline mutation carriers [94,95]. The phase II registration trial included 61 patients with hereditary *VHL*-associated RCCs; thirty (49%) of these patients demonstrated an objective response by the RECIST criteria, and another thirty subjects experienced stable disease. The median duration of response was not reached within the follow-up period of 21.8 months. Belzutifan also demonstrated significant activity against hemangioblastomas and pancreatic lesions, which were observed among the patients included in the study [96].

There are some recurrent hypomorphic alleles of the *VHL* gene, which render no increased cancer risk to their carriers, but are associated with an abnormally increased erythrocyte count when present in a homozygous or compound-heterozygous state. These alleles have been discovered during the studies of “Chuvash polycythemia”. Chuvashia is an autonomy located near the Volga River whose population managed to preserve significant national identity over centuries. It was revealed in the 1970s that the Chuvash population is characterized by a high frequency of erythrocytosis with a significant trend to familial clustering [20,118,119,120]. The genetic linkage analysis assigned its inheritance to the locus 3p25, and the sequencing of the *VHL* gene revealed a recurrent Chuvash mutation *VHL* p.R200W [121]. “Chuvash polycythemia” was subsequently identified in patients belonging to other ethnic groups with either *VHL* p.R200W or other hypomorphic alleles involved in the disease causation [122,123]. The history of the discovery of “Chuvash polycythemia” underscores the value of ethnicity-specific genetic studies.

### 2.2. Fumarate Hydratase Tumor Predisposition Syndrome (FHTPS)

FHTPS is a new name for hereditary leiomyomatosis and renal cell carcinoma (HLRCC) syndrome [72]. FHTPS/HLRCC was initially described in the 1970s, characterized by the accumulation of cutaneous and uterine leiomyomas (Reed syndrome) [124]. Its association with RCC was noticed only at the beginning of the 21st century [125]. FHTPS/HLRCC is caused by heterozygous germline inactivating mutations in the fumarate hydratase (*FH*) gene [126]. The population incidence of FHTPS/HLRCC is believed to be approximately 1:200,000 [127]. *FH* pathogenic variants are represented by loss-of-function missense mutations, protein-truncating alterations, and, less frequently, deletions of large portions of the *FH* gene; however, there are no established correlations between the type of the mutation and clinical presentation of the disease [23,26]. The amino acid substitutions in codon 190 (R190H, R190L, R190C) are hot-spot mutations, which have been described in distinct populations and result in the dominant negative effect [128]. Some communities are characterized by the persistence of founder *FH* pathogenic alleles [57].

While heterozygous inactivation of *FH* is associated with FHTPS/HLRCC, individuals with biallelic germline FH deficiency (constitutional fumarase deficiency) demonstrate a mitochondrial encephalopathy phenotype but no increased cancer risk. Rare hypomorphic *FH* variants are described, which are associated with FH deficiency while being in a biallelic state but not with HLRCC when present in heterozygotes. The most-studied example is the *FH* c.1431_1433dupAAA (p.Lys477dup) allele [129]. It causes constitutional FH deficiency when present in trans with another loss-of-function *FH* mutation, but its heterozygous carriers are not affected by FHTPS/HLRCC.

Tumor development in heterozygous FH mutation carriers involves somatic inactivation of the remaining copy of the gene. There are also sporadic phenocopies of FHTPS/HLRCC-related tumors, which acquired FH deficiency during the process of malignant transformation [97]. Inherited *FH* heterozygosity results in almost complete penetrance toward leiomyomas, while the lifetime probability of the development of RCC is estimated to be within 10–35% [127]. The data on low population incidence and RCC-specific penetrance [127] are in apparent conflict with the results of a multigene sequencing study, which revealed *FH* pathogenic or likely pathogenic variants in 7/254 (2.8%) RCC patients [9]. There are several other reports suggesting that germline *FH* gene alterations make a noticeable contribution to RCC morbidity [8,12,14,23,130].

FH deficiency inhibits the enzymatic conversion of fumarate to malate. Fumarate may act as an oncometabolite. It inhibits the activity of prolyl hydroxylases, which mediate the degradation of hypoxia-inducible factors. Consequently, *FH* inactivation results in a pseudohypoxic phenotype, particularly in HIF accumulation. There are other biochemical consequences of FH deficiency, i.e., activation of aerobic glycolysis, epithelial-to-mesenchymal transition, an increase in the concentration of reactive oxygen species (ROS), changes in genome methylation, inhibition of double-strand DNA repair by homologous recombination, etc. [57,92,127,131,132,133].

FH-deficient tumors are often described as having papillary morphology, although real-world patient series demonstrate significant diversity in RCC histological presentation [134]. A high level of fumarate results in the succination of cellular proteins. The presence of S-(2-succino)-cysteine residues can be detected by immunohistochemistry and serves as a marker of tumor FH deficiency along with the alterations of FH expression and *FH* gene mutations [97,127]. FH-deficient tumors produce succinyl-adenosine and succinic-cysteine, which can be detected in plasma and potentially serve as a non-invasive marker for early disease diagnosis [135]. FHTPS/HLRCC-related RCCs are highly aggressive; therefore, they need to be considered for surgical removal immediately after diagnosis. Systemic treatment of FH-deficient RCCs usually relies on antiangiogenic drugs, similarly to sporadic RCCs [97,98]. HIF upregulation results in transcriptional activation of several cancer-driving genes, including epidermal growth factor receptor (EGFR). A combination of bevacizumab and erlotinib has been successfully utilized in clinical trials involving patients with FHTPS/HLRCC-related RCCs [99]. Several studies performed a systematic comparison of the efficacy of different drug combinations in patients with FH-deficient RCCs [97,98]. *FH* inactivation may interfere with antitumor immune response [100,101]. Some studies demonstrated a high efficacy of immune therapy in patients with FH-deficient carcinomas [100,102,103,104,105]; however, other reports showed a low benefit from the immune checkpoint blockade in this clinical setting [97,98]. A combination of multikinase inhibitors with anti-PD1 therapy showed superior results as compared to other treatment options in a retrospective multicenter study [136].

### 2.3. Kidney Tumors Associated with Hereditary Paraganglioma and Pheochromocytoma (HPP) Syndrome

HPP is a rare syndrome attributed to heterozygous germline mutations affecting the components of the succinate dehydrogenase (SDH) complex [27,74]. This enzyme consists of SDHA, SDHB, SDHC, and SDHD subunits. HPP-associated germline mutations have been described for each of the above-mentioned genes as well as for the SDH assembly factor 2 (*SDHAF2*), *MAX,* and *TMEM127* genes [74]. The spectrum of pathogenic alleles is diverse [31], although founder pathogenic variants have been reported in the Netherlands, Northern Europe, Poland, Portugal, Spain, French Canada, South America, etc. [11,28,61,63,64]. The risks of RCC are elevated mainly in carriers of *SDHB* pathogenic alleles; however, even in these subjects, the probability of RCC development is only three times higher than in the general population [29]. The role of the other above-mentioned genes in RCC predisposition remains unproven [72]. The overall contribution of HPP-related alleles in kidney cancer incidence is below 1% [8,9,12,14]. Tumors arising in *SDHx*-heterozygous individuals are characterized by somatic inactivation of the remaining gene copy. SDH deficiency prevents the conversion of succinate to fumarate; hence, HPP- and HLRCC-related carcinomas have significant similarities [74,131]. SDH-deficient RCCs are highly aggressive and require immediate surgical removal when detected at an early stage. The rarity of HPP-associated RCCs complicates the clinical assessment of therapeutic options for this category of kidney tumors [71,74,110,137].

### 2.4. Hereditary Papillary Renal Cell Carcinoma (HPRCC)

HPRCC is a familial cancer syndrome with autosomal-dominant inheritance [71,110]. While the majority of familial tumors are caused by the transmission of an inactivating mutation in a tumor suppressor gene and require somatic inactivation of the remaining gene copy to trigger the process of malignant transformation, HPRCC is related to activating mutations in the *MET* oncogene. In this respect, HPRCC is similar to multiple endocrine neoplasia types 2 and 3 (also known as type 2B), which are attributed to the inherited activation of another receptor tyrosine kinase, *RET* [2]. *MET* activation is involved in the pathogenesis of several tumor types with the most well-known example being exon 14 skipping mutations in lung cancer [138]. Surprisingly, *MET*-driven cancer-predisposing syndrome is manifested only by papillary kidney tumors and no extrarenal disease manifestations [71,110]. All *MET* pathogenic alleles are missense variants [21,139,140,141]. The population incidence of HPRCC is unknown, indicating that this is an exceptionally rare disease. *MET* pathogenic alleles are detected in approximately 0.4% of patients with RCC, while their frequency in subjects with type 1 papillary renal carcinomas approaches one out of eight [9,21,110]. *MET* mutations are observed in approximately two out of five subjects reporting a family history of this disease [23]. *MET*-associated HPRCCs usually emerge during adulthood with the life-long disease penetrance achieving almost 100% [71]. Sporadic papillary RCCs often carry somatic activating alterations in the *MET* oncogene; therefore, they are essentially similar in their pathogenesis to HPRCCs [71,140].

Low-sized HPRCCs do not have high metastatic potential; therefore, the “watch-and-wait” strategy is usually applied to tumors with a diameter below 3 cm [71,110]. Organ-sparing tumor resection is recommended for cancers exceeding this threshold. The systemic therapy for *MET*-driven tumors is relatively straightforward, given the availability of a few highly potent and specific MET tyrosine kinase inhibitors. Capmatinib has been recently approved for the treatment of *MET*-driven lung cancer; however, its trials on HPRCC are compromised by the rarity of this disease [142]. Crizotinib is available in most countries across the world for the treatment of *ALK*- or *ROS1*-driven lung carcinomas [138]. This drug also has activity towards MET kinase, and it demonstrated a promise in several trials or case reports involving papillary RCC patients, including some subjects with *MET*-mutated tumors; however, no clear-cut distinction has been made between hereditary and sporadic RCCs [107,108,109]. The HPRCC literature often refers to a successful clinical trial of the dual MET/VEGFR2 inhibitor foretinib, which involved 10 patients with germline *MET* mutations, despite the fact that the clinical development of this drug has been discontinued [143].

### 2.5. Kidney Tumors Associated with Birt–Hogg–Dubé (BHD) Syndrome

The current definition of BHD syndrome refers to a hereditary disease, which is characterized by cutaneous hamartomas, pulmonary cysts resulting in pneumothoraxes, and renal tumors. Historically, this syndrome was initially described by Otto P. Hornstein and Monika Knickenberg in the year 1975 in two siblings, and was characterized by the accumulation of perifollicular fibromas and some other disease manifestations [144]. Two years later, Arthur R. Birt, Georgina R. Hogg, and W. James Dubé independently described a large kindred with the inheritance of similar skin lesions [145]. The recognition of other features of BHD syndrome, i.e., lung cysts and kidney carcinomas, emerged years later after the initial description of this disease [146]. While skin abnormalities constitute the most recognizable feature of BHD in patients of the European race, the manifestation of BHD in Asian subjects often does not involve signs detectable by visual medical examination [147,148].

Genetic studies revealed that BHD is caused by loss-of-function heterozygous mutations in the folliculin (*FLCN*) gene [149]. *FLCN* pathogenic alleles are represented by protein-truncating genetic lesions, inactivating missense variants, and large deletions. There is a mutation hotspot located within the track of eight cytosines [146]. In addition, some communities are characterized by the persistence of founder BHD-predisposing variants [33,67,68]. *FLCN* pathogenic alleles have incomplete penetrance, as the frequency of PVs in the population is approximately an order of magnitude higher than the number of clinically diagnosed BHD patients [31,34]. Renal tumors develop in approximately one out of five BHD patients and demonstrate somatic inactivation of the remaining gene copy [36,146]. The overall contribution of *FLCN* PVs is approximately at or below 1% [8,12,14]. They are more often represented by hybrid oncocytic tumors or chromophobe RCCs, although other histological appearances may be observed as well. In contrast to hereditary malignancies, sporadic chromophobe RCCs are not characterized by *FLCN* gene lesions. *FLCN*-related RCCs are usually not aggressive; therefore, the “watch-and-wait” strategy is utilized for tumors with a size below 3 cm, followed by surgery for larger malignancies. Only a few instances of metastatic spread of renal cancer have been described for BHD patients [146]. *FLCN* inactivation results in the upregulation of the mTOR signaling pathway. The experience of using the mTOR inhibitor everolimus in patients with BHD is limited to one case report published in the Japanese language in which the administration of this drug resulted in longer progression-free survival as compared to previously utilized sorafenib and sunitinib [111].

### 2.6. Tuberous Sclerosis

Tuberous sclerosis (TS) is a well-known autosomal-dominant genetic disease with an incidence exceeding 1:10,000 [39,40,150]. It manifests with multiple benign tumors affecting the brain, heart, kidney, and other organs. Episodes of epilepsy and cognitive alterations are highly characteristic for TS patients [151]. The disease is caused by loss-of-function alterations in the *TSC1* and *TSC2* genes. Germline alterations in *TSC1* and *TSC2* can be detected in approximately 90% of patients with clinical signs of TS, while the remaining subjects are assigned to the no-mutation-identified (NMI) category [150,152]. Some NMI patients develop their disease due to mosaic alterations of the *TSC1/2* genes, although the majority of NMI cases remain unexplained [153]. TS patients often have severely affected quality of life and social adaptation; therefore, they have reduced chances to find a spouse and to have children. Consequently, more than two-thirds of identified *TSC1/2* mutations are de novo events [150,152]. A significant share of TS-associated genetic alterations is represented by gross deletions affecting mainly the *TSC2* gene; therefore, patients with negative results of NGS analysis should be subjected to MLPA testing [152,154].

Renal involvement in TS patients is manifested by renal cysts and angiomyolipomas [150]. *TSC2* deletions often extend to the neighboring gene, *PKD1*, leading to the simultaneous manifestation of polycystic kidney disease [6,115,155]. *TSC1/2* mutation carriers have a several-fold increased risk of RCCs [115]. The contribution of *TSC1/2* PVs in RCC incidence is below 1% [9,12,13]. These carcinomas are characterized by significant diversity in their histological appearance [115,116]. *TSC1/2* gene inactivation results in the upregulation of the mTOR pathway. Expectedly, mTOR inhibitors have shown high clinical efficacy toward TS-associated tumors [112,113,114].

### 2.7. Other Hereditary Cancer Syndromes Associated with Increased Risk of Kidney Tumors

*PTEN* hamartoma tumor syndrome (PHTS), which includes Cowden syndrome and some other conditions, is associated with the development of benign and malignant tumors affecting the breast, thyroid, endometrium, kidney, and other organs. Its association with renal cancer risk is well-established, although quantitative estimates vary by an order of magnitude [48,156,157].

The *BAP1* gene plays a role in the regulation of DNA repair, transcription, programmed cell death, and mitochondrial metabolism. *BAP1*-associated tumor predisposition syndrome is a rare disease that is primarily associated with mesothelioma and uveal melanoma. Approximately 7% of *BAP1* PV carriers develop RCCs [158]. Both familial and de novo instances of mutations have been described [53,70]. Occasional *BAP1* PV carriers are reported in all series of RCC patients subjected to germline multigene testing [8,9,12,13]. Despite a small number of observations, *BAP1*-associated RCCs are known to have an aggressive disease course [106,110].

*CHEK2* PVs are particularly common in several countries with predominantly Slavic populations and some regions of Northern and Central Europe. *CHEK2* is well-recognized as a breast cancer-predisposing gene, while its association with other tumor types is less proven [2]. There are several datasets suggesting that *CHEK2* heterozygosity is associated with an increased risk of kidney cancer [71,159,160].

Wilms’ tumor (nephroblastoma) is a relatively common tumor among children. Its most frequent genetic cause is a heterozygous loss-of-function germline mutation affecting the *WT1* gene. This tumor is often accompanied by other clinical signs associated with inherited *WT1* alterations, i.e., aniridia, genitourinary abnormalities, etc. So-called overgrowth syndromes are also associated with an increased risk of nephroblastoma development. Wilms’ tumor may be observed in subjects affected by some other well-known hereditary diseases, e.g., Bloom syndrome, Fanconi anemia, Li-Fraumeni syndrome, *DICER1* syndrome, etc. [161]. *DICER1*-associated syndrome may sometimes manifest by kidney sarcomas [162,163,164].

Current literature often describes the *MITF* p.E318K amino acid substitution as a variant associated with a moderately increased risk of renal cancer. This role has been suggested in the study [165], but subsequent investigations failed to confirm this relationship [166]. Multigene sequencing studies revealed that the frequency of the *MITF* p.E318K variant in RCC patients is similar to the one observed in population databases [8,11].

Lynch syndrome is attributed to germline mutations affecting mismatch repair (MMR) genes and is manifested mainly by colorectal and endometrial carcinomas [2]. Some studies reported an increased risk of renal malignancies in carriers of MMR PVs; however, the available data are contradictory [167,168].

There are a number of “novel” RCC-associated genes: *PRDM10*, *ELOC1/TFEB1*, *NBR1*, etc. However, virtually all relevant reports are generally limited by the description of single pedigrees; therefore, the population-based significance of these findings remains to be clarified [72,169,170,171,172].

## 3. Conclusions and Perspective

Increasing availability of high-throughput sequencing will definitely lead to the discovery of new RCC-predisposing genes in the near future. The analysis of yet unstudied populations is particularly promising, given that each ethnic group has its own ancestors and, therefore, a unique burden of genetic disease. The above-presented example of the discovery of Chuvash polycythemia provides convincing support for this concept [20,118,119,120].

Most of genetic conditions associated with renal malignancies have incomplete penetrance towards renal manifestation. Not surprisingly, multigene sequencing studies of more or less non-selected kidney cancer patients revealed a significantly higher than expected frequency of RCC-associated PVs [8,9,11,13,14]. The analysis of other categories of cancer patients demonstrated a similar tendency [2]. The currently available penetrance estimates were obtained mainly via pedigree analysis. An unbiased comparison of PV frequencies in cancer patients and controls will certainly result in the reconsideration of risks associated with the RCC genes. A search for penetrance-modifying factors is of particular importance, given that at least some carriers of PVs are likely to remain healthy during their lifetime and, therefore, may not require tight medical surveillance.

Hereditary kidney malignancies have specific roots for tumor development and provide an opportunity for molecularly tailored therapies. Indeed, recent years have brought a breakthrough in the systemic treatment of tumors associated with von Hippel–Lindau disease, tuberous sclerosis, and germline *MET* mutations (Table 4) [96,113,143]. FH- and SDH-deficient carcinomas also have unique vulnerabilities and hold promise for the development of targeted therapies [132,137]. There is also significant progress in surgical techniques utilized for the management of hereditary kidney tumors [173].

The development of an efficient approach to early diagnosis, surgical removal, and systemic treatment of kidney tumors has led to a significant increase in life expectancy in carriers of RCC-predisposing mutations [5,174]. The emphasis on the quality of life of these subjects is of particular importance for future studies in this field.

## Figures and Tables

**Figure 1 medsci-12-00012-f001:**
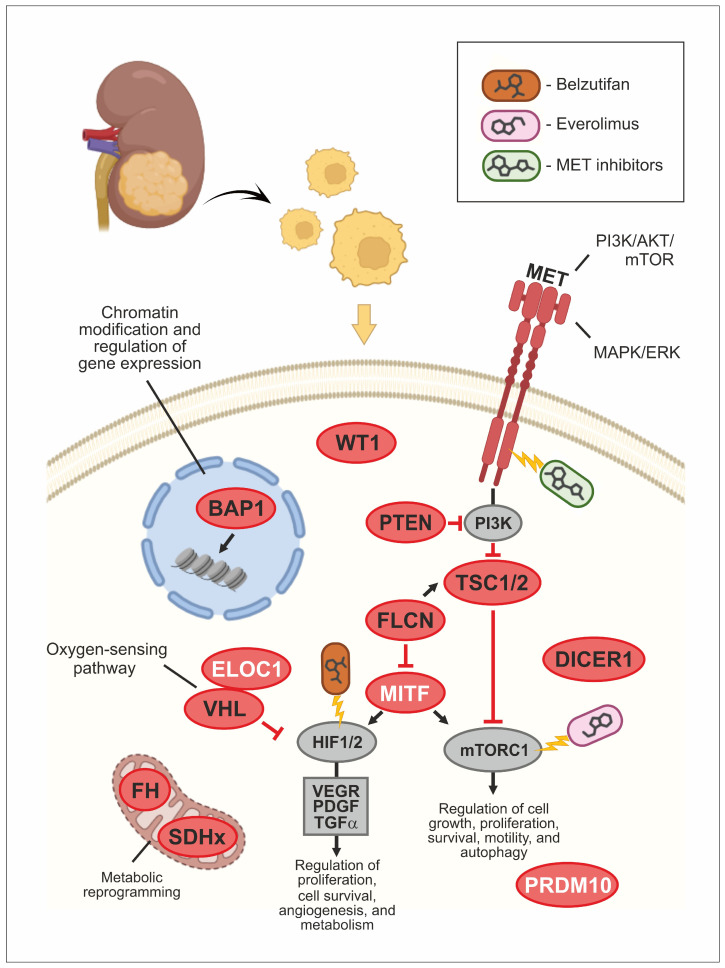
Molecular pathways involved in the development of hereditary kidney tumors and their therapeutic targeting.

**Table 1 medsci-12-00012-t001:** Hereditary syndromes involving kidney tumors.

Disease	Gene	Population Frequency of PVs *	Incidence of the Disease in Population	Contribution to Cancer Morbidity			PVs Penetrance	% De Novo	Comments
				Patients Satisfying Clinical Criteria of the Syndrome	Kidney Tumors **	Non-Kidney Tumors			
Von Hippel–Lindau disease (VHL)	*VHL*	16–32/141,456 (1.13–2.26 × 10^−4^/1:4424–1:8850)	Incidence in newborns: 1:27,000 [5] Prevalence: 1:46,900 (Denmark) [5]; 1:36,000–1:91,000 [6]	90% [8]	0.3–3.2%: 0.3% [7,8]; 0.4% [9]; 0.5% [10,11]; 1.7% [12]; 2.2% [13]; 3.2% [14]	Hemangioblastomas: 25–38% [15] Pancreatic neuroendocrine tumors: 10% [16] Pheochromocytomas/paragangliomas (PPGLs): 4–5% [17,18]	87–95% [5,19] Hypomorphic missense mutations associated with isolated PPGLs may have reduced penetrance	20% [3,5]	Genotype– phenotype correlations are observed. Type 1A disease (reduced risk of PPGL): truncating mutations, inactivating missense mutations; Type 1B disease (reduced risk of PPGL, RCC): large gene rearrangements (LGRs) involving neighboring *BRK1* gene; type 2A disease (reduced risk of RCC), type 2B (high risk for all VHL-associated malignancies), type 2C disease (isolated PPGL): missense mutations. Some mutations are associated with autosomal-recessive congenital erythrocytosis [20]
Hereditary papillary renal cell carcinoma syndrome (HPRCC)	*MET*	4/141,456 (2.83 × 10^−5^/1:35,335)	Unknown (this syndrome is considered to be very rare [7,14,21])	N/A	0.4% [9] Papillary type 1 renal tumors: 12.4% (sporadic: 5%; familial: 41%) [21]	None	~100% [21]	Not reported	
Fumarate hydratase tumor predisposition syndrome (FHTPS) or hereditary leiomyomatosis and renal cell carcinoma (HLRCC)	*FH*	29–35/141,456 (2.05–2.47 × 10^−4^/1:4049–1:4878) ^#^	Unknown Leiomyomas are frequent finding in general population, and *FH* PVs have relatively low penetrance towards kidney cancer	40% (multiple cutaneous leiomyomas and/or multiple uterine fibroids and/or papillary type 2 RCC) [22] 70% (familial HLRCC cases) [23]	0.2–5.2%: 0.2% [11]; 0.6% [13]; 1% [7,10]; 1.6% [14]; 1.8% [11]; 2.8% [9]; 5.2% [12]; Papillary type 2 RCC: 17.4% [23]	Uterine leiomyomas: ~0.7% [24] Uterine leiomyomas in young patients (<30 years): 2% [25]	RCCs: ~19% [26] Cutaneous and uterine leiomyomas: almost complete penetrance Rarely, PPGLs are observed	Not reported	Individuals with biallelic germline *FH* inactivation demonstrate frequently fatal mitochondrial encephalopathy (constitutional fumarase deficiency). Cancer is not a feature of biallelic FH deficiency. Rare hypomorphic variants are described, which are associated with *FH* deficiency but not with HLRCC. *FH*-associated RCCs are very aggressive
Hereditary paraganglioma/ pheochromocytoma (HPP) syndrome	*SDHB, SDHC, SDHD* (paternally inherited), *SDHA*	*SDHB:* 51–53/141,456 (3.6–3.75 × 10^−4^/1:2668–1:2778) ^†^ *SDHD:* 16–51/141,456 (1.13–3.75 × 10^−4^/1:2668–1:8850) *SDHC:* 23–30/141,456 (1.62–2.12 × 10^−4^/1:4717–1:6173) *SDHA:* 136–146/141,456 (9.61 × 10^−4^–1.03 × 10^−3^/1:971–1:1041)	Unknown Prevalence of SDHx-associated PPGLs: 1:51,667–1:77,500 (Denmark) [27]	N/A	0.7–0.9%: 0.8% [9]; 0.7% [11]; 0.8% [14]; 0.9% [8,12]	PPGLs: 20–43% [17,18,28]	PPGLs: *SDHB*: 21.8%; *SDHD:* 43.2%; *SDHC*: 25% [29] *SDHA*: clinical series: 10–30%; population-based estimates: 0.1–4.9% [30] *SDHx*-associated RCCs: *SDHB*: 4.2%; *SDHD*, *SDHC, SDHA*: the increased risk has not been proven [29] Rarely: thyroid carcinomas, GIST, pituitary adenomas, etc. [29]	Not reported	Individuals with biallelic germline *SDHA, SDHB, SDHD* inactivation frequently demonstrate fatal mitochondrial disorders. Cancer is not a feature of biallelic *SDHx* deficiency. Hypomorphic variants are described, which are associated with SDHx deficiency but not with familial PPGL Penetrance of *SDHx* (especially *SDHA*) mutations observed in relatives of PPGL patients is significantly higher than in accidentally identified individuals harboring the same PVs [30]
Birt–Hogg– Dubé syndrome (BHD)	*FLCN*	25–38/141,456 (1.76–2.68 × 10^−4^/1:3731–1:5682)	Varying estimates (this disease seems to be underdiagnosed): 1:500,000 (worldwide) [31] 1:176,366 (South Korea) [32] 1:3265 (Sweden) [33] 1:3234 (Pennsylvania, USA) [34]	67% [35]; 80–85% [35]	0.3–1.6%: 0.3% [11,13]; 0.4% [13]; 0.5% [10]; 1.2% [8]; 1.6% [14]		RCCs: 19–21%; spontaneous pneumothorax/multiple bilateral pleural/subpleural cysts: 82–87%; skin lesions (fibrofolliculomas, acrochordons, angiofibromas): 78–87%; colonic polyps: 21–32% [36]. Population-based study [36]: cystic lung disease: 65.7%; pneumothorax: 17.1%; skin lesions: 8.6%; RCCs: 2.9%	A single report of de novo mutation [37]	In Asian patients, skin lesions are usually subtle and do not raise suspicion in patients or physicians. For example, in one Japanese study, skin lesions were noted in 49% of the patients, but only 1/76 (1.3%) subjects voluntarily consulted a dermatologist before the BHD diagnosis [38]
Tuberous sclerosis (TS)	*TSC1*, *TSC2*	*TSC1:* 5–10/141,456 (3.53 × 10^−5^–7.07 × 10^−5^/1:14,144–1:28,329) *TSC2:* 3–13/141,456 (2.12 × 10^−5^–9.19 × 10^−5^/1:10,881–1:47,170)	~1:10,000 e.g., 1:12,658 (Germany) [39] Incidence in newborns: 1:5800–1:10,000 [40]	75–90% [40]	0.2–2.6%: 0.2% [8]; 0.4% [12]; 0.9% [13]; 2.6% [10]		Cortical tubers: 88–90% (associated with *TSC2*); SEGA (subependymal giant cell astrocytomas): 5–24.4%; cardiac rhabdomyomas: 34–58%; angiomyolipomas: 51.8%; RCCs: 1–2% (unusually early-onset); cystic kidney disease: 50%; lymphangioleiomyomatosis (female patients): 34–81%; angiofibromas: 57.3–74.5%; periungual fibromas: 15%; retinal hamartomas: 30–44% [41]	*TSC1:* 59%; *TSC2*: 85% (more severe phenotype) [42]	
Cowden syndrome	*PTEN*	24–27/141,456 (1.7–1.91 × 10^−4^/1:5236–1:5882)	1:200,000 [43]	~9.5% [44]	0.3% [8]	Breast carcinomas:0.2% [45] Thyroid carcinomas: 0.8% [46] Endometrial carcinomas: <0.4% [47]	Breast carcinomas: 85%; thyroid carcinomas: 35%; kidney carcinomas: 34%; endometrial carcinomas: 28%; other cancers: 9% [43]	10–48% [48]	
*BAP1*-associated tumor predisposition syndrome	*BAP1*	4–48/141,456 (2.82 × 10^−5^–3.39 × 10^−4^/1:2950–1:35,461) ^‡^	Unknown (this syndrome is considered to be very rare [49])	N/A	0.3–1.6%: 0.3% [13]; 0.4% [8,12]; 1.2% [9]; 1.6% [10]	Uveal melanomas: 2–4% (familial uveal melanomas: 22%) [50] Malignant mesotheliomas: 1–5% [51]	Uveal melanomas: 8.5–31%; malignant mesotheliomas: 17–22%; melanocytic *BAP1*-mutated atypical intradermal tumors: 18%; cutaneous melanomas: 3–13%; RCC: 3–10% [52]	Up to 9.5% [53]	

* Allele frequencies are given in accordance with the gnomAD database, version v2.1.1 (https://gnomad.broadinstitute.org/; accessed on 1 November 2023); PVs: pathogenic/likely pathogenic variants, according to ClinVar. ** Eight NGS-based studies were considered: Carlo et al. [9] (254 consecutive patients with advanced RCC from USA; RCC-related genes: *VHL*, *FH*, *FLCN*, *MET*, *SDHB*, *SDHC*, *SDHD*, *BAP1*, *TSC1/2*, *MITF*); Wu et al. [10] (190 early-onset Chinese patients with RCC, incl. 29 angiomyolipomas; RCC-related genes: *VHL*, *TSC1/TSC2*, *PTEN*, *MET*, *FH*, *SDHB*, *SDHC*, *SDHD*, *FLCN*, *BAP1*, *MITF*); Abou Alaiwi et al. [8] (1829 RCC patients from USA; each patient was tested for 1–134 genes); Kong et al. [13] (322 consecutive RCC patients from China; RCC-related genes: *BAP1*, *FLCN*, *FH*, *MET*, *MITF*, *PTEN*, *SDHA*, *SDHB*, *SDHC*, *SDHD*, *TSC1*, *TSC2*, *VHL*); Santos et al. [7] (294 consecutive patients with metastatic RCC and 21 RCC patients with hereditary cancer features from Spain; RCC-related genes: *BAP1*, *FLCN*, *FH*, *MET*, *MITF*, *PTEN*, *SDHA*, *SDHB*, *SDHC*, *SDHD*, *TSC1*, *TSC2*, *VHL*); Truong et al. [12] (232 early-onset RCC patients from USA; 50% had non-clear cell histology; RCC-related genes: *VHL, FH, FLCN*, *MET*, *TSC1/2*, *BAP1*, *SDHA/B/C/D*, *MITF*, *PTEN*); Yngvadottir et al. [11] (1336 consecutive RCC patients from Iceland; the RCC-related genes: *BAP1*, *FLCN*, *FH*, *MET*, *MITF*, *PTEN*, *SDHA*, *SDHB*, *SDHC*, *SDHD, TSC1*, *TSC2, VHL*); Feng et al. [14] (123 consecutive RCC patients from China; RCC-related genes: *BAP1*, *FLCN*, *FH*, *MET*, *MITF*, *PTEN*, *SDHA*, *SDHB*, *SDHC, SDHD*, *TSC1*, *TSC2*, *VHL*). # Two P/LP *FH* missense variants (p.Arg74Lys; p.Ala308Gly) identified in the gnomAD database (1 instance each) are associated with autosomal-recessive fumarase deficiency but not with autosomal-dominant HLRCC. ^†^ 11 instances of *SDHB* missense variant p.Asp48Val were excluded as it is associated with autosomal-recessive mitochondrial complex II deficiency only. ^‡^ *BAP1* p.Tyr401Ter variant (39 instances in gnomAD) is considered pathogenic in ClinVar but actually have not been observed in patients with *BAP1*-related tumor predisposition syndrome. This suggests its reduced penetrance or benign status.

**Table 2 medsci-12-00012-t002:** Mutational spectra in hereditary kidney cancer genes.

Disease	Gene	Spectrum of PVs	Founder Mutations
Von Hippel–Lindau disease	*VHL*	Missense: 52%; frameshifts: 13%; nonsense: 11%; large gene rearrangements (LGRs): 11%; splice-site: 7%; in-frame deletions/insertions: 6% (945 *VHL* families) [54] Recent studies show an increased proportion of LGRs, e.g., 20% in a Japanese data set [55]	Regional founder variant, c.292T>C (p.Tyr98His or c.505T>C, according to old nomenclature), is associated with VHL type 2A disease (also known as “Black Forest mutation” (Germany, Bavaria)) [19]
Hereditary papillary renal cell carcinoma syndrome	*MET*	Activating missense mutations [21]	Not reported
Fumarate hydratase tumor predisposition syndrome	*FH*	Missense: ~50%, nonsense and frameshift mutations: 25–33%. LGRs are rarely described (~5%) [23,26]	Several minor founder variants exist, e.g., Iranian Jewish 905–1G>A mutation (4 families) [56]. There are known hot-spot mutations, e.g., those affecting codon 190 [57]. Recurrent mutations can be identified in the gnomAD database, e.g., c.698G>A (p.Arg233His) variant: 4/50,738 Northwestern Europeans
Hereditary paraganglioma/ pheochromocytoma (HPP) syndrome	*SDHB*, *SDHC*, *SDHD*, *SDHA*	*SDHB*, *SDHC*, *SDHD*: missense: 44%; nonsense: 15%; splice-site: 13%; frameshifts: 15%; in-frame deletions: 0.5%; large CNVs (frequently involve 1st exon/promoter of *SDHB*, *SDHC*, *SDHD*): 12% [29]	Multiple instances of founder mutations are known: Icelandic *SDHA* c.91C>T (p.Arg31Ter) mutation (7/9 (78%) *SDHx*-associated kidney cancer cases); this variant is also frequent in Sweden [13]; Dutch founder mutations in *SDHB*: exon 3 deletion (30% of pathogenic alleles), c.423+1G>A (20% of pathogenic alleles) [28]; Portuguese founder deletion in the 1st exon of *SDHB*: 26–37% of pathogenic *SDHx* alleles in Portuguese PPGL cases [58,59]; the same LGR occurs in Northern Spain [60] and is the most frequent *SDHB* mutation in Brazil (36% mutations) [61] and Colombia (90% mutations) [62]. *SDHC* p.Arg133Ter variant is a French Canadian founder mutation of French origin (69% *SDHx* mutations) [63]; *SDHD* p.Cys11Ter is a Polish founder variant [64]
Birt–Hogg– Dubé syndrome	*FLCN*	The majority of mutations are truncating, e.g., duplications (46.4%), deletions (29.0%), substitutions (7.1%), insertions (0.7%), deletions/insertions (0.3%), large genomic deletions (4.0%), and splice-site mutations (12.5%) (Japan) [65]. Similar results are reported in locus-specific *FLCN* mutation database: deletions (44.3%), substitutions (35.7%), duplications (14.3%), and deletions/insertions (5.7%) [66].	In most populations, a recurrent hot-spot mutation is responsible for up to half of the cases [35]. Several founder variants were reported: Danish founder mutation c.1062+2T>G (11/31, 35% of all mutations) [67]; Chinese regional founder LGR (deletion of exons 1–3) [68]; Swedish founder mutation c.779+1G>T (57% pathogenic alleles) [33]
Tuberous sclerosis	*TSC1*, *TSC2*	*TSC1*: the vast majority of mutations are truncating; *TSC2*: roughly 30% of variants are missense, and 6% are LGRs [40]	Not reported
Cowden syndrome	*PTEN*	Missense mutations: 29%; nonsense mutations: 32%; small deletions: 14%; small insertions: 8%; indels: 1%; large deletions 3%; splice-site mutations: 10%; promoter mutations 3% [44]	Not reported
*BAP1*-associated tumor predisposition syndrome	*BAP1*	~78% are truncating/null variants; 22% are missense mutations; LGRs are rare [49]	There are recurrent hot-spot variants, e.g., p.Arg60Ter. Finnish founder variant (p.G549Vfs*49) has been described [69]. An extremely large cancer family of Swiss origin, scattered across USA, carries p.Leu573fs*3 allele [70]

**Table 3 medsci-12-00012-t003:** Tumor surveillance in patients with hereditary renal cancer syndromes.

Disease	Kidney Tumors	Extrarenal Tumors
Von Hippel–Lindau disease [83]	Biannual MRI since 15 years	Since birth: physical examination; dilated eye examination (every 6–12 months); since 2 years: annual blood pressure and pulse assessment; since 5 years: annual measurement of plasma-free metanephrines; since 11 years: biannual brain and spine MRI and audiogram; 15–20 years: MRI of internal auditory canal (once if no findings detected)
Fumarate hydratase deficiency [84]	Annual MRI since 8–10 years	No specific surveillance
Hereditary pheochromocytoma/ paraganglioma syndrome [85]	MRI every 2–3 years since childhood	Annual blood pressure and pulse assessment; measurement of plasma-free metanephrines or urinary metanephrines (biannually in children, annually in adults); head and neck, thoracic, abdominal, and pelvic MRI (every 2–3 years)
Hereditary papillary renal cell carcinoma [86]	Annual MRI since 30 years	No specific surveillance
Birt–Hogg– Dubé syndrome [87]	Biannual MRI since 20 years	Dermatologic examination (every 6–12 months); annual evaluation of parotid glands; annual ultrasound of thyroid gland may be considered; lung CT for symptomatic patients (before scheduled general anesthesia or before long-distance flights); regular colonoscopies, especially in kindreds with family history of colorectal malignancies
Tuberous sclerosis [88]	MRI every 1–3 years since childhood; annual renal function assessment	Brain MRI every 1–3 years; in asymptomatic infants: electroencephalography (every 6 weeks before 1 year, then every 3 months until 2 years old); electrocardiography every 3–5 years; echocardiography every 1–3 years in asymptomatic patients until regression of cardiac rhabdomyomas; annual dermatologic and ophthalmic examination. Lung CT (every 5–7 years) and annual pulmonary function assessment in adult females
*BAP1* tumor predisposition syndrome [89]	Annual MRI since 30 years	Annual dermatological examination; periodic ophthalmological examination
Cowden syndrome [90]	Biannual ultrasound since 40 years	Since 18 years: annual thyroid ultrasound; since 30 years: annual breast MRI; dermatologic examination (once, at time of diagnosis); Since 35–40 years: colonoscopy (once, at time of diagnosis); Since 40 years: annual mammography

Abbreviations: MRI, magnetic resonance imaging; CT, computed tomography.

**Table 4 medsci-12-00012-t004:** Specific treatment options for hereditary kidney cancers.

Disease	Therapy	Surgical Approach
Von Hippel–Lindau disease	Antiangiogenic therapy and multikinase inhibitors;	“Watch-and-wait” approach (lesions < 3 cm in diameter) [82]
	Anti-HIF2-alpha therapy (belzutifan) [93,94,95,96]	
Fumarate hydratase deficiency	Bevacizumab + erlotinib; multikinase inhibitors; immune therapy [97,98,99,100,101,102,103,104,105]	Immediate surgical removal [106]
Hereditary pheochromocytoma/ paraganglioma syndrome		Immediate surgical removal [72]
Hereditary papillary renal cell carcinoma	MET inhibitors (crizotinib, capmatinib) [107,108,109]	“Watch-and-wait” approach (lesions < 3 cm in diameter) [71,110]
Birt–Hogg– Dubé Syndrome	mTOR inhibitors (everolimus)? [111]	“Watch-and-wait” approach (lesions < 3 cm in diameter) [106]
Tuberous sclerosis	mTOR inhibitors (everolimus) [112,113,114]	“Watch-and-wait” approach for angiomyolipomas, insufficient evidence regarding RCC [115,116]
*BAP1* tumor predisposition syndrome		Insufficient evidence, but immediate surgical removal is suggested [106]
Cowden syndrome	mTOR inhibitors (everolimus)? [117]	Insufficient evidence

## Data Availability

No new data was created or analyzed in this study. Data sharing is not applicable to this article.

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
