# Peer review of "Hereditary Renal Cancer Syndromes"

_medsci, 2024, doi:10.3390/medsci12010012_

Round 1

Reviewer 1 Report

Comments and Suggestions for Authors

I read the manuscript titled "Hereditary kidney cancer" with interest. I suggest the following title "Hereditary Renal Cancer Syndromes".

I suggest you use MeSH terms for keywords.

In general, although this is not the first narrative review on this topic, I think you have put in the effort.

Please, at the end of the introduction, briefly state the purpose of the review. I also ask that the tables and figures appear within the text in the order you list them.

I suggest that you form section 2 under the title diseases/syndromes or similar according to your opinion and that you form all diseases/syndromes as subsections (2.1., 2.2., etc...).

Are you the authors of figure 1?

It is not usual to list a table within the conclusion.

More than 50% of the references are from the last 8 years, which is commendable.

Comments on the Quality of English Language

Moderate editing of English language required.

Author Response

Comment: I suggest you use MeSH terms for keywords.

Response: We have changed the keywords.

Comment: Please, at the end of the introduction, briefly state the purpose of the review. I also ask that the tables and figures appear within the text in the order you list them.

Response: We have inserted the following statement: “This review provides an update on clinical and genetic aspects of inherited predisposition to kidney cancer”.

Comment: I suggest that you form section 2 under the title diseases/syndromes or similar according to your opinion and that you form all diseases/syndromes as subsections (2.1., 2.2., etc...).

Response: This is done.

Comment: Are you the authors of figure 1?

Response: Yes

Comment: It is not usual to list a table within the conclusion.

Response: We have moved this table close to the first mention on the treatment of hereditary cancer syndromes.

Reviewer 2 Report

Comments and Suggestions for Authors

The authors present a nicely written review of the various hereditary kidney cancers.  The review is comprehensive, up to date, and contains useful tables.

A few areas of improvement

1. the surveillance recommendations of these different syndromes should be listed somewhere (e.g. what imaging modality, how often, and at what age to start).  Table 3 is an option

2. The controversy around FH c.1431_1433dupAAA (p.Lys477dup) should be mentioned with respect to HLRCC and FH deficiency

3. the pathway figure 1 could be improved in terms of being more professional looking.  This could be done in collaboration with a medical graphic artist.

Comments on the Quality of English Language

no comments, english is adequate

Author Response

Comment: The surveillance recommendations of these different syndromes should be listed somewhere (e.g. what imaging modality, how often, and at what age to start).  Table 3 is an option

Response: We have inserted a new table describing the surveillance recommendations.

Comment: The controversy around FH c.1431_1433dupAAA (p.Lys477dup) should be mentioned with respect to HLRCC and FH deficiency

Response: We have inserted comments on this issue:

“While heterozygous inactivation of FH is associated with FHTPS/HLRCC, individuals with biallelic germline FH deficiency (constitutional fumarase deficiency) demonstrate a mitochondrial encephalopathy phenotype but no increased cancer risk. Rare hypomorphic FH variants are described, which are associated with FH deficiency while being in a biallelic state, but not with HLRCC when present in heterozygotes. The most studied example is the FH c.1431_1433dupAAA (p.Lys477dup) allele [129]. It causes constitutional FH deficiency when present in trans with another loss-of-function FH mutation, but its heterozygous carriers are not affected by FHTPS/HLRCC.”

Comment: The pathway figure 1 could be improved in terms of being more professional looking.  This could be done in collaboration with a medical graphic artist.

Response: We have revised this figure.